# A Distributed Black-Box Adversarial Attack Based on Multi-Group Particle Swarm Optimization

**DOI:** 10.3390/s20247158

**Published:** 2020-12-14

**Authors:** Naufal Suryanto, Hyoeun Kang, Yongsu Kim, Youngyeo Yun, Harashta Tatimma Larasati, Howon Kim

**Affiliations:** School of Computer Science and Engineering, Pusan National University, Busan 609735, Korea; naufalso@pusan.ac.kr (N.S.); hyoeun0915@pusan.ac.kr (H.K.); dkgoggog0329@pusan.ac.kr (Y.K.); yeo8006@pusan.ac.kr (Y.Y.); harashta@pusan.ac.kr (H.T.L.)

**Keywords:** adversarial examples, particle swarm optimization, distributed attack

## Abstract

Adversarial attack techniques in deep learning have been studied extensively due to its stealthiness to human eyes and potentially dangerous consequences when applied to real-life applications. However, current attack methods in black-box settings mainly employ a large number of queries for crafting their adversarial examples, hence making them very likely to be detected and responded by the target system (e.g., artificial intelligence (AI) service provider) due to its high traffic volume. A recent proposal able to address the large query problem utilizes a gradient-free approach based on Particle Swarm Optimization (PSO) algorithm. Unfortunately, this original approach tends to have a low attack success rate, possibly due to the model’s difficulty of escaping local optima. This obstacle can be overcome by employing a multi-group approach for PSO algorithm, by which the PSO particles can be redistributed, preventing them from being trapped in local optima. In this paper, we present a black-box adversarial attack which can significantly increase the success rate of PSO-based attack while maintaining a low number of query by launching the attack in a distributed manner. Attacks are executed from multiple nodes, disseminating queries among the nodes, hence reducing the possibility of being recognized by the target system while also increasing scalability. Furthermore, we utilize Multi-Group PSO with Random Redistribution (MGRR-PSO) for perturbation generation, performing better than the original approach against local optima, thus achieving a higher success rate. Additionally, we propose to efficiently remove excessive perturbation (i.e, perturbation pruning) by utilizing again the MGRR-PSO rather than a standard iterative method as used in the original approach. We perform five different experiments: comparing our attack’s performance with existing algorithms, testing in high-dimensional space in ImageNet dataset, examining our hyperparameters (i.e., particle size, number of clients, search boundary), and testing on real digital attack to Google Cloud Vision. Our attack proves to obtain a 100% success rate on MNIST and CIFAR-10 datasets and able to successfully fool Google Cloud Vision as a proof of the real digital attack by maintaining a lower query and wide applicability.

## 1. Introduction

Over the years, the use of deep learning has been found in the broad range of applications to perform various tasks, ranging from image classification, object recognition, and social network analysis [1], to name a few. Due to its remarkable development in image prediction ability, deep learning has also been considered as one of the key technologies in autonomous vehicles, enabling the car to recognize traffic signs, such as stop, speed limitation, and lane change. As the use-cases continue to grow, research on vulnerability and threats posed by deep learning also gains attention. One of the most intriguing vulnerabilities that have been found to date is the adversarial example, in which seemingly similar images in human eyes are perceived differently by classifiers in deep learning models. This susceptibility is the base of adversarial attacks, in which a small perturbation (i.e., carefully crafted "noise") imperceptible to human vision are added to an input image, causing performance degradation of the model, and even result in image misclassification [2].

Earlier methods for generating a perturbed image for adversarial attacks mainly run under the white-box settings, which is less applicable in the real-world applications. In this approach, adversaries have the information of the target model’s structure, parameter, training dataset, or even the learned weight. Since the adversaries own detailed knowledge about the inner of the target model [3], an adversarial example can be crafted easily. However, in real-life scenarios, there are not many cases of an attacker having an internal view of a deep learning model. Hence, current methods are more towards the black-box settings, in which an attacker should be able to deceive the model with the constraint of using only predicted values for input data without knowledge of the internal model.

Unfortunately, many proposed techniques under black-box settings employ a large number of queries to the model in order to successfully generate adversarial examples. This large queries problem is because in black-box settings, typical attack strategies are performed by probing the system: inputting perturbed inputs to the target system until the ones able to fool the system are found, which may be achieved after thousands of queries. When implementing this attack, for instance, to artificial intelligence (AI) service providers (e.g., Google Vision), the large number of queries coming from attacker node is likely to be detected when the AI providers have been aware of this attack. Another type of attack method is by utilizing a substitute model and transferring from the white box settings, which is computationally exhaustive, hence making it inefficient and not favorable in a time-limit constraint.

A recent method that offers time efficiency and a much lower query in creating adversarial examples is the AdversarialPSO [4], which utilizes Particle Swarm Optimization (PSO). PSO is a gradient-free, machine-learning optimization algorithm which performs a heuristic search by certain mechanism inspired by the way a flock of birds navigates when searching for food. Despite its superiority to other methods, AdversarialPSO incurs a relatively low attack success rate for a high dimension image. This is probably due to the nature of PSO, which aims to quickly find a relatively good solution (i.e., local optima) rather than a global solution. Hence, once it falls into a local optimum, it will be difficult to continue the search for a better solution.

Nevertheless, we believe that those weaknesses can be minimized, mainly by utilizing two mechanisms. Firstly, by implementing the attack in a distributed manner. Borrowing the example from network security perspective, it is known that the Distributed Denial of Service (DDoS attack, i.e., attacking a web server by sending massive numbers of requests which is scattered from multiple nodes) are much harder to detect and overcome rather than its single-node counterpart, the Denial of Service (DoS) attack. This characteristic can also apply to our approach for adversarial attacks. By disseminating the queries to multiple nodes, the number of queries perceived by the AI provider will be much lower, hence lessening the possibility of being detected. Additionally, employing a distributed approach means that the computational resource required to perform the attack can be dispersed into several nodes, lowering the device specification requirement. Using a distributed approach, it will be possible to execute attacks from low computational hardware, such as the Raspberry Pi cluster.

Secondly, by employing the multi-group version of PSO algorithm for adversarial example generation, namely the Multi-Group Particle Swarm Optimization with Random Redistribution (MGRR-PSO) [5]. MGRR-PSO is a modification of standard PSO which can outperform the standard PSO in terms of finding the global optima. This algorithm, which has been proposed on the standard case of optimization function (not adversarial example), reaches higher maxima and lower minima on all tested benchmark functions. MGRR-PSO combines two groups of PSO with opposite acceleration coefficients and enables redistribution to reach global optima and avoid being stranded in local optima. Incorporating MGRR-PSO for perturbation generation is a promising solution to improve the current PSO-based adversarial attack. By performing these two mechanisms, the opportunity to yield lower query to avoid detection while maintaining a high attack success rate can be achieved.

In this paper, we propose a distributed black-box adversarial attack based on MGRR-PSO. In particular, the total number of queries are distributed to each client to generate the adversarial example, making it harder to be detected by AI service providers. Furthermore, we utilize the MGRR-PSO, which we prove in our experiment, performs better than AdversarialPSO in terms of attack success rate. Our method, inspired by Reference [4], consists of two steps: perturbation generation for creating the adversarial example, and perturbation pruning to remove the nonessential perturbation, creating an adversarial image very close to the original image. Additionally, we propose to improve the efficiency of current methods to remove excess perturbation (i.e., perturbation pruning) by again, utilizing MGRR-PSO rather than the standard iterative method. Additionally, by employing a black-box setting, our method is applicable for actual attack scenarios, such as a malicious user who uses AI service remotely with little knowledge of the target model. To verify our method, we test it on the MNIST, CIFAR-10, and ImageNet dataset and then compare its performance to several recently proposed methods.

Distributed queries for adversarial examples have earlier been mentioned in Reference [6] in Section 8, in which the authors acknowledge that distributed queries will remain hard to detect. Apart from that, there is no further discussion about the design and implementation of such approach. To the best of our knowledge, the implementation of adversarial examples generation in a distributed manner has not been found in the existing literature. Furthermore, we are the first to employ a multi-group variant of Particle Swarm Optimization (PSO), which, in the standard optimization case, is proven to be effective in reaching global maxima, for generating adversarial examples. Our contributions can be summarized as follows:We design and implement a distributed architecture for adversarial attacks in black-box settings to reduce the number of queries which, to the best of our knowledge, has never been proposed in the existing literature.We apply a multi-group variant of Particle Swarm Optimization (PSO) algorithm called Multi-Group PSO with Random Redistribution (MGRR-PSO) for adversarial example generation, which outperforms the current PSO-based adversarial attack algorithm in terms of escaping local optima, hence yielding higher attack success rate.We propose a more efficient removal technique of unnecessary perturbation in the generated adversarial example by also utilizing MGRR-PSO.We demonstrate that by employing a distributed architecture and utilizing MGRR-PSO for adversarial attacks, we can achieve high success rates and a low number of queries on each attacker node in various scenarios.

## 2. Background and Related Work

In this section, we firstly introduce adversarial examples and types of typical attack methods. Consequently, we discuss black-box adversarial attacks and related work employing this attack method. Finally, we review Particle Swarm Optimization (PSO) and its variant, which can be utilized for generating adversarial examples.

### 2.1. Adversarial Examples and Attack Methods

An adversarial example is a modified version of a clean image that is intentionally perturbed to trick a machine learning model [7] into making a false prediction. For generating adversarial examples, there are two main types of attack methods: white-box attack and black-box attack.

In a white-box attack, the attacker has the perfect knowledge of the target model [8]. In other words, the attacker is aware of the information related to the targeted model, such as architecture, weights, and parameters. Szegedy et al. [9] was the first to introduce adversarial examples against deep neural networks, in which they utilize the L-BFGS method. However, their method is time-consuming and impractical due to the expensive use of linear search method to find the optimal value [2]. Goodfellow et al. [10] proposed an efficient method called Fast Gradient Sign Method (FGSM) to generate adversarial examples. They argued that neural networks are too linear to resist linear adversarial perturbation. They suggested a fast linear method to attack non-linear neural networks that uses the gradient of the loss function, which can be computed efficiently. Papernot et al. [11] proposed Jacobian-based Saliency Map Attack (JSMA), which utilized a saliency map to perturb specific features and generate adversarial examples efficiently. Deepfool [12] is an untargeted attack technique optimized for the $L_{2}$ distance metric. It is based on an iterative linearization of the neural networks to generate optimal perturbations with a minimal size that is sufficient to attack the classifier. Additionally, Carlini and Wagner [13] considered a targeted attack as an optimization problem. The attack generates adversarial examples with a minimal difference, similar to the one presented in Reference [9]. This attack is found to be effective against most of existing adversarial detecting defenses [2], primarily the defensive distillation [13].

In a black-box attack, the attacker only has little knowledge of the target model, even sometimes without knowing the labeled data. A black-box attack with constraints often consists of fewer queries to the model and the predicted confidence scores of the model. This approach is closer to the real-world application because, in reality, deep learning models used in various application fields are commonly protected by a security system; hence, it is difficult to extract information, such as model structures and parameters. Since the limited access of the black-box attack to the system reflects the real-world problem [3], it is considered more practical compared to the white-box attack. To further explore this more-applicable attack, we provide an elaboration of previous works that utilize black-box adversarial attacks against the classification model in the following subsection.

### 2.2. Black-Box Adversarial Attacks

#### 2.2.1. Common Approaches

In the black-box attack, the attacker needs to craft adversarial inputs with minimal capability. There are some techniques commonly used for generating adversarial examples: transferability, gradient estimation, and gradient-free approach. Transferability concept allows adversarial attack by approximation using a substitute model. In the case of gradient estimation, black-box attacks estimate the model gradient to create adversarial examples, quite similar to those employed in white-box attacks that use gradient information. For gradient-free approach, the attack strategy mainly is by querying perturbed inputs to the target model and observing the resulting output and then modifying the perturbed input to yield a better solution by some cost function.

In transferability approach, one of the representative techniques is the method proposed in Reference [6]. In the paper, Papernot et al. proposed substitute model training for creating adversarial examples. In the process of generating a substitute model, the attacker collects data regarding the input domain and selects the model architecture associated with the input domain. The attacker repeats the process of learning and augmenting the collected data so that they can better represent the target model. Jacobian-based dataset augmentation is used to iteratively refining the substitute model. Consequently, the attacker can carry out white-box attacks against a substitute model to emulate the target model. An ensemble of local models [14] is employed rather than just a single local model. For determining the loss of the adversarial example, the sum of the loss of several models is taken into account instead of one single model.

In the context of gradient estimation approach, Chen et al. [15] proposed Zeroth Order Optimization (ZOO) to estimate gradients of the target model. This approach uses a derivative-free method by evaluating objective function values at two very close points. Furthermore, this attack evaluates coordinates after adding a small perturbation to estimate the direction of the gradient for each coordinate instead of all coordinate updates when image size is large. Another method in this approach is by Tu et al. [16], who proposed Autoencoder-based Zeroth Order Optimization Method (AutoZOOM) for query-efficient crafting of adversarial examples. AutoZOOM takes an adaptive random gradient estimation strategy for adjusting the trade-offs between estimation error and query counts. AutoZOOM requires a decoder in order to add a perturbation in high dimension from latent-space in low dimension.

In the case of gradient-free approach, Narodytska et al. [8] proposed a greedy local search for creating adversarial examples. This strategy does not require the information of the target model’s gradient. After selecting a subset of points (called local neighborhood) to be perturbed and evaluating every point, the attacker calculates a new solution using the previous solution, as well as the results of evaluated points. If the acquired solution is not an adversarial example, the attacker will update to new points which is close to the current points. The evaluation is repeated until a valid adversarial example is created. In this method, it is possible to attack by changing only a few pixel values in the image. Another approach is by utilizing the genetic algorithm, a population-based gradient-free optimization strategy, as proposed by Alzantot et al. [17]. Genetic algorithm is inspired by natural selection and gradually evolves a population of candidate solutions towards an optimal solution. The process of this attack is as follows: (1) generating random samples; (2) filtering subset of samples by a proper fitness function; (3) producing next solution through crossover which mix remaining samples in (2); (4) mutating samples to add small perturbation by user-defined probability to increase diversity. This process is repeated until adversarial examples are found.

#### 2.2.2. Limitations of Current Approaches

Each approach has its own downsides. Black-box attacks using gradients derived from a substitute model (i.e., the transferability approach) require the step of training the model, which adds to the overhead in terms of time required to launch an attack. Furthermore, the adversarial example applied in the substitute model may not be transferred effectively to the target model. In terms of gradient-based approach, the attacks can fail if the obtained gradient is caught in local optima. A number of queries could enable the attacker to obtain an adequate gradient estimation to avoid local optima, but at the expense of a considerable amount of time, so that AI providers can quickly respond to the attack. Furthermore, calculating the gradient requires significant computational resources to craft adversarial examples. Gradient estimation methods are well adjusted to find global optima, which requires more queries to craft adversarial examples. In terms of gradient-free approach, training is not required since the attack is typically built heuristically. This technique is found to converge quickly while maintaining a good quality solution, hence reducing the number of queries to the target model. However, it does not guarantee a global optimal solution due to its trait to quickly converge, sometimes trapped to local optima, which can result in a low success rate.

#### 2.2.3. Evaluation Metric for Black-Box Attacks

The primary metric for black-box attacks is the attack success rate, which is the rate of misclassification when the generated adversarial example is inputted to the target model. Another metric is average distortion, which is the difference between an adversarial example and the original image. The lower the average distortion of an adversarial example, the harder it is for a person to notice the difference between the two images. Additionally, in query-based attacks, such as gradient estimation and gradient-free approach, the number of queries also has to be measured. Roughly, the number of queries corresponds to the time required to form an adversarial image. A low query attack that achieves a high attack success rate means that the attack can be carried out more effectively and fast, hence becoming harder to be detected by the AI provider. Therefore, it is favorable that a black-box attack has a high attack success rate, low average distortion (e.g., $L_{2}$ and $L_{\infty }$ distance), and low queries.

### 2.3. Particle Swarm Optimization (PSO) for Adversarial Attacks

#### 2.3.1. Particle Swarm Optimization Overview

Particle Swarm Optimization (PSO) [18] is a gradient-free optimization algorithm inspired by the social behavior of moving organisms, such as the movement of a bird flock, to achieve the same goal. The algorithm optimizes a problem (i.e., cost function) by iteratively attempting to improve candidate solutions (called particles) based on previous knowledge of each candidate and entire populations (called swarm). The particles move around the search space according to simple formulas. The movement is based on calculated velocity influenced by the particle’s best-known position (i.e., local best) in the search space, as well as the best-known position of the entire swarm (i.e., global best). Suppose the search space has *d*-dimension (d=1,2,…,D), the candidate positions and velocities of each particle *i* are represented by $X_{i}=\left(x_{id},\ldots ,x_{iD}\right)$ and $V_{i}=\left(v_{id},\ldots ,v_{iD}\right)$, respectively. For each iteration or time-step *t*, the velocities are used to update the next positions of each particle calculated as:(1)$$v_{id}\left(t+1\right)=wv_{id}\left(t\right)+c_{1}r_{1}\left(p_{gd}-x_{id}\left(t\right)\right)+c_{2}r_{2}\left(p_{id}-x_{id}\left(t\right)\right),$$
(2)$$x_{id}\left(t+1\right)=x_{id}\left(t\right)+v_{id}\left(t+1\right).$$

For each particle *i* and dimension *d*, update the particle’s velocity $v_{id}$ in Equation (Equation 1) and particle’s position $x_{i}d$ in Equation (Equation 2) through the above equations. In Equation (Equation 1), *w* is the momentum weight impacting on how much of the previous velocity affect the next particle movement. The swarm’s best position is denoted as $p_{gd}$ and is weighted with the constant $c_{1}$, which is positive acceleration coefficient of global best. Similarly, the best position of each particle $p_{id}$ is weighted with the positive acceleration coefficient of local best, which is denoted as $c_{2}$. $r_{1}$ and $r_{2}$ are uniform random number sample from U(0,1).

#### 2.3.2. AdversarialPSO

Most of the previous works on adversarial examples leverage gradient-based optimization, which tends to fall into local minima. Nevertheless, a white-box attack commonly suits this approach due to the fact that the target model structures and parameters can be procured by the attacker. On the other hand, Mosli et al. [4] proposed AdversarialPSO, which utilizes Particle Swarm Optimization (PSO), a gradient-free optimization technique, with some modifications to craft adversarial examples. In AdversarialPSO, particles’ movement creates perturbations that are incredibly small that humans cannot perceive them in the image. In a black-box environment, PSO is very suitable for crafting adversarial examples since a few queries are found to generate practical adversarial examples, making it efficient. However, the attack success rate is relatively lower than other proposed methods.

#### 2.3.3. MGRR-PSO

Multi-Group Particle Swarm Optimization with Random Redistribution (MGRR-PSO) [5] is an algorithm that leverages a multi-group approach, in which it combines two groups of PSO with opposite acceleration coefficients. In MGRR-PSO, there are two groups of PSO that have opposite acceleration coefficients of $c_{1}$ and $c_{2}$ in Equation (Equation 1). The first group has bigger $c_{1}$ than $c_{2}$, and the other group has bigger $c_{2}$ than $c_{1}$. In other words, the first group moves to the global best stronger, and the other group moves more to the local best. Equations (Equation 3) and (Equation 4) describe how to assign $c_{1}$ and $c_{2}$ with MGRR-PSO’s acceleration coefficients A1 and A2.
(3)$$c1=\left\{\begin{array}{cc} max(A1,A2) & \mathrm{if}i<particle\_size/2,\\ min(A1,A2) & \mathrm{otherwise} \end{array}\right.,$$
(4)$$c2=\left\{\begin{array}{cc} min(A1,A2) & \mathrm{if}i<particle\_size/2,\\ max(A1,A2) & \mathrm{otherwise} \end{array}\right..$$

Both groups work together and share their global best position. The combination of these two groups makes more variation in searching the optimized solution around the search space. In the work of Reference [5], which is unrelated to adversarial attacks, the authors have shown that MGRR-PSO overcomes the weakness of standard PSO that the particles often easily trapped into local optima. Every time the particles are trapped in the same position, half of the first group particles and half of the second group particles are randomly redistributed. When the redistributed particles search a new global best position, all particles will move to that position. This movement eases the particles to escape local optima.

## 3. Overview of Proposed Attack

### 3.1. Attack Description

Our proposed black-box attack is the distributed adversarial attack based on Particle Swarm Optimization (PSO) algorithm variant that is scalable, harder to detect by the AI service provider, and with a high attack success rate. Rather than utilizing a single client, this attack uses multiple attacker clients that connect to an attacker server, working together to generate an adversarial example, as illustrated in Figure 1a. Each client act as several particles running the multi-group version of PSO, namely the MGRR-PSO algorithm. Each particle represents the possible solution of an adversarial example. Clients query the generated adversarial example from each particle to the target AI service provider to obtain the confidence score of the desired class for further evaluation. Clients also inform their search results to the server so that the current best adversarial example is acquired and become a reference as the global best for the particles on other clients. The total number of queries to generate an adversarial example is distributed to all clients, yielding a much fewer query per client, hence making it harder to be noticed by the target AI service provider. The more number of attack clients exists, the faster and the lesser required queries on each client for generating adversarial example. The sequence diagram of this attack is as shown in Figure 1b.

There are two main steps that we employ for generating adversarial examples with minimal perturbations. The first step is the adversarial example generation, in which we utilize MGRR-PSO. The second step is the perturbation pruning, which is an additional step to remove unnecessary perturbation to further reduce the visibility of adversarial example, by again using MGRR-PSO. In our distributed architecture, each client runs these steps independently from other clients, with periodic communication to the server for updating the global best. Since we use multiple clients, parallel execution yields a faster time to create an adversarial example.

### 3.2. Image Representation

In our study, the goal is to find the best adversarial image by utilizing a heuristic multi-group particle swarm optimization. We represent the value of each pixel in the image *x* as particle position $x_{i}$. Since an image is represented in a 3-D array with height *H*, width *W*, and depth *D*, we first transform it into a 1-D array. Hence, the resulting particle position will have H×W×D values. Let *h*, *w*, and *d* as the pixel position on height, width, and depth index, respectively. The mapping of a 3-D image into a 1-D position array can be written as:(5)(h,w,d)↦(h+H·(w+(d·W))).

To revert the particle position into a 3-D image array, the mapping can be written as:(6)(i)↦(imodH,(i/H)modW,i/(H·W)).

For the remainder of this paper, discussion related to particle position will refer to the 1-D position with index *i* instead of the 3-D image array counterpart.

### 3.3. Fitness Function for Untargeted and Targeted Attack

In this proposal, we implement multi-group particle swarm optimization with certain fitness function *J* to measure and evaluate the success of each particle in searching the adversarial example. We define two fitness functions: for untargeted attack and targeted attack, respectively.

In untargeted attacks, the goal of a fitness function is to minimize the confidence score of the original class, while, on targeted attacks, the goal is to maximize the confidence score of the target class. Let $f_{o}\left(x^{\prime }\right)$ and $f_{t}\left(x^{\prime }\right)$ be the model’s confidence score in the original and target label when predicting the adversarial example $x^{\prime }$, respectively. We can simply write the fitness function of an untargeted attack as:(7)$$J\left(x^{\prime }\right)=f_{o}\left(x^{\prime }\right).$$

To convert the goal of a targeted attack into the same minimization problem, we can write the fitness function of a targeted attack as:(8)$$J\left(x^{\prime }\right)=1.0-f_{t}\left(x^{\prime }\right),$$
where the value of $f_{o}\left(x^{\prime }\right)$ and $f_{t}\left(x^{\prime }\right)$ is between 0.0 and 1.0.

For untargeted attacks, by minimizing the probability of the original label, the probability of other labels will increase. When the other label’s probability becomes higher than the original label, the image will be misclassified. On the other hand, for targeted attacks, by maximizing the probability of the target label, other labels’ probability will decrease until the target label becomes highest; hence, images will be misclassified as the label that we have targeted.

In AdversarialPSO [4], the authors introduce a mean squared error (MSE) $L_{2}$ penalty term to their proposed fitness function. It is expected to promote smaller perturbations when producing adversarial examples. However, as shown in the figures in Reference [4], this penalty does not adequately reduce the perturbation in the crafted image (i.e., the perturbation is evident to human vision). In other words, it does not entirely yield efficient reduction. In fact, this penalty slows down the perturbation generation process and reduces the attack success rate for the algorithm to find adversarial examples when tested in our experiment if the parameters are not finely tuned. For these reasons, we do not employ $L_{2}$ penalty term on our adversarial example generation.

### 3.4. Suppressing Perturbation Visibility

To narrow the visibility of generated perturbation, we employ the perturbation limit mechanism, which is similar to that of AdversarialPSO [4], described by the following equation.
(9)$$x_{i}^{\prime }\left(t+1\right)=min(0,max\left(1,\left(x_{i}^{\prime }\left(t\right)+v_{i}\left(t+1\right),x_{i}^{\prime }-B,x_{i}^{\prime }+B\right)\right).$$

We set the search boundary *B* that limits maximum perturbation value or $L_{\infty }$ distance. $L_{\infty }$ distance measures the maximum difference between the original image *x* and adversarial image $x^{\prime }$ on every pixel. We use the min and max operators (i.e., clip) to limit the search boundary every time the particle updates its position. In addition, we keep the value of the adversarial image into a valid image range value between 0.0 and 1.0.

## 4. Distributed Black-Box Adversarial Attack Using MGRR-PSO

In this section, we describe our proposed method in detail. Specifically, we explain the hyper-parameters defined at the initialization stage of the attack and elaborate the steps of adversarial example generation, as well as perturbation pruning, employed in our technique.

### 4.1. Attack Initialization

#### 4.1.1. MGRR-PSO Hyper-Parameter

In the PSO-based technique, we need to define the particle size $P_{size}$. For MGRR-PSO itself, particle size should be the multiple of two and consist of at least four because the particles will be divided into two sub-groups. Additionally, we also need to define *w*, *A1*, and *A2*, which are the parameter values of inertia weight and two different acceleration coefficients, respectively. *A1* and *A2* are used to assign local best acceleration coefficient *c1* and global best acceleration coefficient *c2* for each sub-groups, as shown in Algorithm 1.

Original MGRR-PSO as proposed in Reference [5] redistributes half of its particle on a fixed half random dimension when the optimization is converged. However, we find that for a very high dimension input (i.e., image), randomizing half of the total pixel is inefficient and may slow down the search process. Hence, we introduce a new hyper-parameter rand_rate for MGRR-PSO to determine the random number of particle dimension that will be redistributed instead of using fixed half value. This improves the variety of particle movements across the search space.

There is one thing to consider regarding the particles. In general knowledge, employing larger $P_{size}$ could deliver a better solution (i.e., corresponds to a higher attack success rate). However, larger $P_{size}$ corresponds to larger number of query in one iteration (i.e., in one update), $Q_{i}$, despite not necessarily corresponding to larger required total number of query to reach convergence, $Q_{tot}$. In other words, there is a trade-off between $P_{size}$ (corresponds to attack success rate) and $Q_{i}$ (number of query). Please note that this trait is in the context of PSO-based algorithm alone, regardless of the deployed architecture (distributed or single node).

#### 4.1.2. Distributed Architecture Hyper-Parameter

Since we employ a distributed architecture, firstly, we need to define the number of attacker clients $k_{size}$ that will run the MGRR-PSO algorithm (i.e., how many clients to launch the distributed attack). In this attack architecture, it is preferable to use the number of particle as little as possible to minimize the number of queries on each client. To boost speed and attack success rate, we can maximize $k_{size}$, enabling the attack to run in parallel. In our technique, each client $k_{i}$ may have a different value of those constants to enable different particle behavior in crafting the adversarial example.

**Algorithm 1:****MGRR-PSO Attack**—for cost function *J*, position of particle pos, best position of each particle pBest, best position of all particle gBest, coefficient of local best c1, coefficient of global best c2, velocity of particle *v*, random value *r*

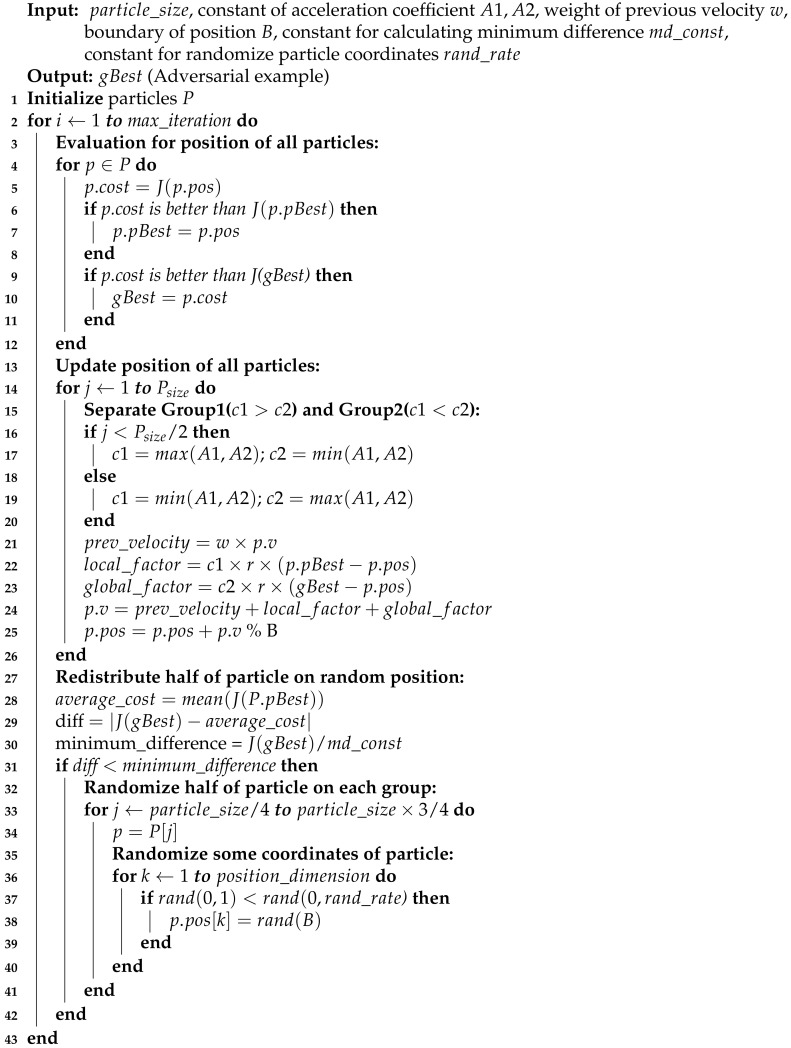



#### 4.1.3. Attack Hyper-Parameter

Before launching the adversarial attack, we need to define the search boundary *B*, which will also be the maximum distance between the generated perturbed image and the original image, $L_{\infty }$. *B* determines the quality of the generated adversarial example. Lower *B* will produce an adversarial example that is less perceptible by humans. Unfortunately, finding an adversarial example using a low $L_{\infty }$ value may require more queries. To limit the number of queries, we can set the maximum iteration of MGRR-PSO and define the early stop criteria. If the maximum iteration is reached while the stop criteria is not met, we can restart the searching by increasing the value of the search boundary to enable wider search space. It is tolerable to have high $L_{\infty }$ at the beginning because the proposed perturbation pruning algorithm will reduce the number and the visibility of generated perturbations.

### 4.2. Adversarial Example Generation

This step is an iterative process for the particle *P* in each client *k* to move for finding better fitness value $J(x^{\prime })$. Particle’s position *pos_i_* represents the value of image pixel on coordinate *i*. This position is updated based on calculated velocity *v*, which comes from three vectors: previous positions, particle’s local best position *pBest*, and global best position *gBest* stored in the attacker server. The position update is limited to the search boundary *B* so that the difference between original image *x* and adversarial image $x^{\prime }$ is under some visibility threshold. On every iteration, each particle evaluates its position using predefined fitness function for either the untargeted attack or the targeted attack. This step requires each particle to perform a query, inputting its position $x_{i}^{\prime }$ to the target AI provider, for acquiring confidence scores which will be its fitness value $J(x^{\prime })$. If the acquired fitness value is better than its current personal best, it will update its own personal best *pBest*. In addition, it will update the global best solution *gBest* if the fitness is better than the global best. Additionally, half of the particle in each subgroup will be redistributed when the average *pBest* is close to *gBest* to avoid premature convergence. This process is repeated until the maximum iteration is reached, or depending on the predefined stop criteria (e.g., image is successfully misclassified, or the program reaches a specific value in confidence scores) is met. Algorithm 1 describes the detailed flow of MGRR-PSO that we use to generate an adversarial example.

### 4.3. Perturbation Pruning

A generated adversarial example often produces redundant perturbation that supposedly can still be minimized. AdversarialPSO [4] introduces a simple iteration method for reducing excess perturbation from an adversarial image by probing along perturbed pixel and multiply that perturbation value by half. While the reduced pixel still yields the same result as the desired class, the reduced image will be the new adversarial image, and this process is performed again iteratively. If the reduced pixel causes the image to return to the original label, the last change will be undone. However, this process is inefficient since an adversary will need to query the image for every perturbed pixel changed. In our approach, we introduce an efficient method for reducing the distance between original and adversarial image by once again, utilizing the MGRR-PSO algorithm, in which we refer to as the perturbation pruning. For this mechanism, we define a new conditional fitness function that makes the particles minimize the $L_{2}$ distance between original and the generated adversarial image under the condition that the result is still unaltered from our desired class. For untargeted attack, we can write the fitness function as:(10)$$J\left(x^{\prime }\right)=\left\{\begin{array}{cc} \left|\right|x^{\prime }-x{\left|\right|}_{2}, & \mathrm{if}\;\mathrm{predicted}\;\mathrm{label}\neq \mathrm{original}\;\mathrm{label},\\ \infty & \mathrm{otherwise} \end{array}\right..$$

For targeted attack, the fitness function is:(11)$$J\left(x^{\prime }\right)=\left\{\begin{array}{cc} \left|\right|x^{\prime }-x{\left|\right|}_{2}, & \mathrm{if}\;\mathrm{predicted}\;\mathrm{label}=\mathrm{target}\;\mathrm{label},\\ \infty & \mathrm{otherwise} \end{array}\right..$$

Additionally, we utilize a new search boundary, which is the negative and positive values of the added perturbation. To extract these negative perturbation *np* and positive perturbation *pp* values, we take the difference between the generated adversarial image $x^{\prime }$ and original image *x* and then trim the values such that:(12)$$np=min(-1,max\left(0,x^{\prime }-x\right)),$$
(13)$$pp=min(0,max\left(1,x^{\prime }-x\right)).$$

These negative and positive perturbation value becomes the new lower and upper search boundary *B = (np, pp)* of perturbation pruning method. We update *B* for each iteration to suppress the perturbation visibility at the pixel level. Using this new search boundary, we can write the new perturbation limit equation as:(14)$$x_{i}^{\prime }\left(t+1\right)=min(0,max\left(1,\left(x_{i}^{\prime }\left(t\right)+v_{i}\left(t+1\right),x_{i}^{\prime }+np,x_{i}^{\prime }+pp\right)\right).$$

The implementation of these new fitness functions and search boundary to MGRR-PSO will result in an efficient perturbation pruning method. By running this method until reaching our defined maximum iteration, we are able to obtain a working adversarial example with minimal perturbation. Figure 2 represents the comparison of the original, adversarial, and pruned image. We can see that the pruned image (i.e., generated adversarial image that undergoes perturbation pruning) incurs much less perturbation.

## 5. Evaluation

### 5.1. Experiment Overview

To assess our proposed technique, we conduct a total of five different experiments as follows. Firstly, to obtain the performance comparison to the existing adversarial attack methods, we evaluate MGRR-PSO attack along with five other methods: Zeroth Order Optimization (ZOO) [15], Carlini and Wagner’s (C&W) black-box attack [13], as well as AdversarialPSO and SWISS [4], on MNIST [19] and CIFAR-10 [20] datasets. Secondly, to investigate how our model perform in high-dimensional space, we also test it on ImageNet dataset using a pre-trained Inception-v3 network. In these two experiments, we employ only one client for launching the attack rather than using our distributed approach (i.e., using more than one client) to give a fairer comparison with other attacks.

On the third experiment, we vary the particle size in one client to see how it affects the required number of iteration, queries, and the average $L_{2}$. In the fourth experiment, we evaluate the scalability of our attack. We run our proposed distributed approach and evaluate the effect of adding more clients for reducing the number of queries and increasing the attack success rate. Specifically, we vary the number of client from one to five. Additionally, the fourth experiment also evaluates the effect of different constrained search boundary *B* with respect to the same evaluation metrics. Furthermore, for the third and fourth experiments, we also apply a simple parametric test (i.e., linear regression) to analyze how strong the relationship between our hyper-parameters and evaluation metrics are. On the last experiment, we test the applicability of our attack in a real digital black-box attack scenario, in which we launch our distributed attack to generate adversarial examples on Google Cloud Vision API [21].

### 5.2. Experiment Setup

#### 5.2.1. Attack Performance Comparison

We conduct an untargeted attack and targeted attack on 1000 and 100 correctly classified samples from test sets, respectively. In the targeted attack, our evaluation is to misclassify the original image into all other classes. We measure metrics of success rate, average $L_{2}$ distance, and average queries. We take the evaluation results of other attack methods from Reference [4]. Since other attacks are presented on non-distributed approach, we run MGRR-PSO attack with only single client for a fairer performance comparison. We employ the same MGRR-PSO’s hyperparameters for all target model: 0.75 for inertia weight *w*, 1.0 for acceleration coefficient *A1*, and 2.0 for acceleration coefficient *A2*, 25 for *md_const*, 0.5 for *rand_rate*, and 500 for the maximum iteration.

On MNIST dataset, we choose 0.15 and 0.2 as the initial search boundary *B* for untargeted attack and targeted attack, respectively. On CIFAR-10, we choose 0.025 and 0.05 as the initial *B* for untargeted attack and targeted attack, respectively. If the maximum iteration is reached while the adversarial example has not been successfully generated, we restart the search process and increment the *B* by 0.025. We use the lowest working search boundary with limited iteration on each test images. In our algorithm, we mark the evaluation as a fail if adversarial example cannot be found until the maximum *B* is reached. This evaluation only uses adversarial example generation algorithm without perturbation pruning.

#### 5.2.2. High Dimensional Evaluation on ImageNet

We conduct the attack experiment for 100 correctly classified test samples on Inception-v3 network of the ImageNet dataset. The hyper-parameters setup of MGRR-PSO are same as in MNIST and CIFAR-10. Additionally, we determine the search boundary *B* 0.05 as the constraint and set the maximum iteration as 3000. We measure the success rate and average queries needed for untargeted attack in the constraint setting.

#### 5.2.3. Hyper-Parameter Evaluation: Particle Size

For particle size evaluation, we mimic the set of the first experiment except for the particle size. We use 4, 6, 8, and 10 particles within a single client.

#### 5.2.4. Hyper-Parameter Evaluation: Distributed Approach and Constraint Search Boundary

For this experiment, we utilize our own evaluation model which we train by ourselves rather than the C&W model to ensure that the performance of the evaluated adversarial attacks algorithm is not tailored only to one specific model. In other words, we also want to verify the validity of our attack. Additionally, our own trained model yields a relatively higher accuracy compared to the C&W’s model; hence, we can compare our result with the state-of-the-art in a more proper way.

We evaluate the effect of adding more client in order to reduce the number of queries and increase the attack success rate. We also evaluate the effect of different constraint search boundary *B* respected to the same evaluation metrics. Furthermore, we employ the attack with different number of client, ranging from 1 to 5, with every client’s number of particle is 4. In addition, we try different number of constraint *B* on each dataset. For MNIST, we use 0.25, 0.2, and 0.15, while, for CIFAR-10, we use 0.1, 0.05, and 0.025. Our attacks are executed in parallel for each client and particles. Since each client works parallelly and asynchronously, it may have different total queries compared to other clients. Hence, we measure the average queries required on each client rather than the total queries.

Our distributed black-box attack environment consists of an attack server and multiple clients that communicate with a simple TCP client/server architecture on a local network. The data sent between parties are in JSON format. If the stop criteria (i.e., image being misclassified) or maximum iteration is reached, each client returns the result to the attacker server.

Regarding the statistical parametric test, we perform a simple linear regression to see the relation between the number of clients and the size of constraint boundary with the resulting average query and success rate.

#### 5.2.5. Real Digital Black-Box Attack on Google Cloud Vision

For this attack, we choose random images on ImageNet dataset and employ the attack in both untargeted and targeted attack. Since Google Cloud Vision model uses multi-label classification, our goal for untargeted attack is to minimize the highest original label until it is missing from the prediction list. For targeted attack, our goal is to maximize our target label probability until its score become the highest. We try different hyper-parameters until our attack is success.

### 5.3. Experiment Result

Overall, the results of our experiments show that our proposed attack yields higher performance than other tested algorithms on both average distortions *L*_2_ and number of queries, as tested on MNIST and CIFAR-10 datasets. Furthermore, we achieve a 100% success rate on both untargeted and targeted attack. Additionally, the experiment in high-dimensional space on ImageNet dataset also yields a relatively high success rate. The next two experiments give some insights for the best practice on this attack. Finally, we prove that our proposed technique is applicable in real digital scenario by successfully executing the attack on Google Cloud Vision model. In the following subsections, we elaborate on the result of each experiment.

#### 5.3.1. Result of Performance Comparison

As presented in Table 1, our method achieves 100% attack success rate of all cases while maintaining a relatively low queries even with only a single client. Among the four scenarios (i.e., untargeted and untargeted attack on MNIST and CIFAR-10), our method yields the lowest query in all cases except in untargeted attack on MNIST dataset. In addition, the average $L_{2}$ value is not relatively high compared to other methods. Hence, adversarial examples generated by our method can be considered as imperceptible. Note that the presented results are without undergoing perturbation pruning; thus, $L_{2}$ value can still be minimized to produce more imperceptible adversarial example. Figure 3 represents the adversarial example results for the MNIST and CIFAR-10 datasets.

#### 5.3.2. Result of High Dimensional Evaluation

Table 2 represents the performance of our method in the untargeted attack setting. Unlike the previous experiment, we cannot achieve 100% attack success rate because for this experiment, we set the constraint for search boundary *B* value to 0.05 and maximum iteration of 3000. Without the constraint setting, the attack success rate can be very close, or even equal to 100%. The reason why we are confident about our result is because during the run of this experiment, the confidence score of original label of adversarial example generated by our method continuously decreases at every iteration. Therefore, if we do not set the constraint setting, a 100% attack success rate could surely be achieved. Figure 4 shows the example result of our generated adversarial examples on ImageNet dataset with *B* set to 0.05. As shown in the figure, the adversarial images can not be easily distinguished from the original images.

#### 5.3.3. Result of Particle Size Evaluation

Figure 5 shows the result of varying particle size on a single client. From Figure 5a, it can be inferred that increasing the particle size can slightly reduce the required number of iteration, but yield a multi-fold increase on the total query, as presented in Figure 5b. Specifically, from the statistical test using linear regression, increasing the number of particle by 1 results in the average number of iteration reduced by 8 and 6 for MNIST and CIFAR-10, respectively, but with the tradeoff of an increasing average query of 238 and 107, respectively.

The reason why the number of queries greatly increases is because the total queries are equal to the number of iteration multiplied by particle size. Therefore, increasing the number of particle is generally not favorable, except for the case that the AI service provider supports batch query operation. In this latter case, the number of query is equal to the number of iteration, thus increasing particle size is advantageous.

Another insight that can be acquired from Figure 5c is the higher the particle size, the lower the $L_{2}$ distance of the generated adversarial example. In particular, the average $L_{2}$ distance is reduced by 0.049 and 0.031 in our test. This results in a cleaner image for the same number of iteration.

#### 5.3.4. Result of Distributed Approach and Constrained Boundary

Figure 6 presents the result of varying the number of clients and the value of constraint boundary *B*. The graphs prove that increasing the number of clients reduces the number of queries on each client, as illustrated in Figure 6a,b. From our test on the MNIST dataset, increasing one client for the boundary of 0.15, 0.2, and 0.25 reduces the average query by 200, 134, and 65, respectively.

Furthermore, with the same number of maximum iteration and boundary value, increasing the number of clients will also increase the success rate, as shown in Figure 6c,d. For the boundary value as mentioned earlier, in the MNIST dataset, increasing one client will increase the average success rate by 2.43, 0.678, and 0.06 percent, respectively.

Additionally, as can be inferred from Figure 6, our experiment shows that a larger boundary yields a smaller average query and higher success rate, but reducing the quality of the generated adversarial examples. Hence, selecting the appropriate constraint value for boundary *B* is essential.

#### 5.3.5. Result on Google Cloud Vision

Figure 7 presents two examples of our adversarial examples generated on Google Vision, which misclassifies a traffic light and a cat as a tree. As shown, the perturbation is visually imperceptible. We have tried with different random images and the algorithm also works well if we can choose the proper search boundary *B* and maximum iteration. For targeted attack, the algorithm works better if we choose the target label that is close to, or exist on the image. By minimizing the original label or increasing the target label probability, the overall predicted label probabilities are changed, yielding different classification result.

## 6. Discussion

**Observations regarding the search boundary *B*.** From the performed experiments, crafting adversarial example using larger search boundary *B* is easier, in the context that it requires fewer queries than that of using lower *B*, thus reducing the possibility of being detected by the target AI service provider. However, this will also cause more distortions on the generated adversarial example, hence increasing the perturbation visibility to humans. Since this is a tradeoff, a proportional *B* value should be considered. Another phenomenon that we observe about our algorithm during our experiment is that it is easier (i.e., use lower query) to find an adversarial example with low *B* in high dimensional image compared with that of low dimensional image. For instance, using a fixed *B* value and a maximum number of queries for all datasets (e.g., *B* of 0.05 and max query of 1000), an adversarial example can be successfully generated in CIFAR-10 and Image-Net (higher dimension), but unsuccessful for MNIST dataset. For MNIST, we will need to increase the *B* value.

**Limitations of our algorithm.** The algorithm works under the condition that we can obtain either the probability of original image (for untargeted attack), or the probability of target image (for targeted attack). Thus, our attack method does not apply to the circumstance when the AI service provider only returns class without its probability (label or one-hot encoding).

**How to keep undetected.** Even though using less queries for each client (as used in our algorithm) can reduce the possibility of detection, running an attack by flooding the target AI service provider with requests in a short duration time can also be a flag. The system can recognize it as an uncommon or anomaly event. To anticipate this risk, we can add a random interval per query for camouflage, thus resembling a normal situation.

**Potential detection mechanism against our proposed attack –and how to dodge it.** Our attack performs queries of relatively similar images over the time to craft the best adversarial examples. As an AI service provider, one of the methods to anticipate this kind of attack is by logging the clients’ requests and check whether the current query has a small distance image with the previous image. If largely similar images are queried within some period, it is very likely that an adversary is structuring an attack on that provider’s deployed model. Fortunately for the adversary, this detection scenario can still be eluded. Since our attack is distributed, different attacker clients can take turns to perform the query sequentially.

**Opportunity for scalability.** The distributed approach may enable a more scalable attack, in the context that for launching a computationally-expensive attack, we can simply add another computing resource with the same specification to reduce the computing power requirement by roughly a half. However, since PSO is a computationally low algorithm, it actually does not need to be scaled up. For another case, we can add any commodity hardware, such as a laptop, to join the attack network to distribute the computing load. Furthermore, in the era of the widely-deployed thin clients (e.g., tablets, mobile phones) and the Internet of Things (IoT), like now and the future ahead, it is not impossible to deploy hundreds or thousands of clients from different places to generate adversarial examples. Then, the adversarial attack can be executed in only one step if the number of clients is huge enough. Not only adversarial example generation in black-box settings will be faster than ever, but it will also be awfully hard to recognize this attack.

**Potential use of other PSO-based algorithms in distributed attacks.** From the mathematical optimization perspective, adversarial example generation can be considered as a combinatorial optimization problem since the solution representation is in a discrete form. Our proposed attack leverages the use of MGRR-PSO algorithm, which intrinsically uses continuous representation as in the standard PSO. Nevertheless, there are other kinds of PSO-based algorithms that are tailored for discrete representation, such as the one proposed by Santucci et al. [22], namely Algebraic PSO, which tackles the permutation-based problems using a completely discrete representation. It is interesting to investigate how our attack expands to other domains, which will be an intriguing future research question.

## 7. Conclusions

In this paper, we propose a black-box adversarial attack using Multi-Group Particle Swarm Optimization with Random Redistribution (MGRR-PSO), which is less detectable by the target system. We distribute the total number of queries to multiple attacker clients, lowering the number of queries from each node, hence reducing the possibility of attack detection from the target system. Additionally, we employ MGRR-PSO algorithm, which can escape local optima for adversarial example generation, resulting in a high success rate. Additionally, we propose a method to remove unnecessary perturbation in the generated adversarial example by also leveraging the MGRR-PSO algorithm, reducing the image distance, hence making it more imperceptible to humans. Our experiment results on the MNIST, CIFAR-10, and ImageNet datasets with various image classification models show that our attack method is applicable and can generate effective adversarial examples with very high success rate under the black-box setting while maintaining a low number of query to the target model. Using the combination of distributed attack and utilizing MGRR-PSO algorithm, we achieve a less detectable system while maintaining a high success rate compared to the state-of-the-art.

## Figures and Tables

**Figure 1 sensors-20-07158-f001:**
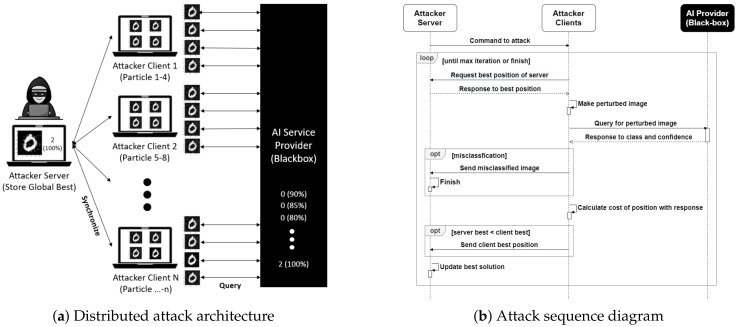
Architecture overview of distributed black-box adversarial attack using Multi-Group Particle Swarm Optimization with Random Redistribution (MGRR-PSO).

**Figure 2 sensors-20-07158-f002:**
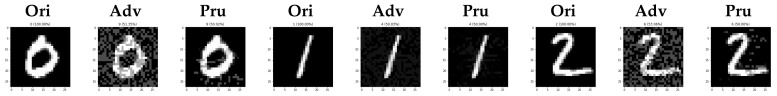
Comparison of original, adversarial, and pruned image: by applying perturbation pruning algorithm to the generated adversarial example, we can remove the inessential perturbation.

**Figure 3 sensors-20-07158-f003:**
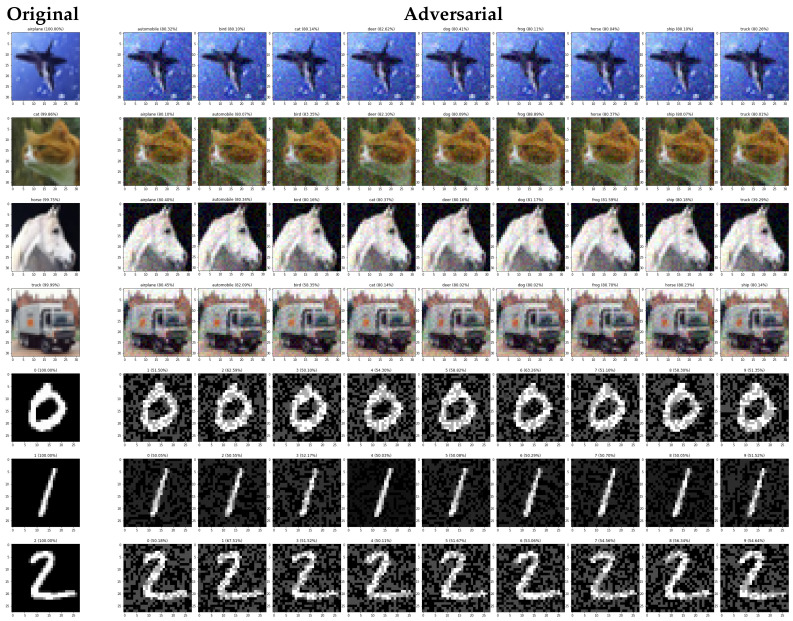
Our adversarial examples on CIFAR-10 and MNIST datasets: The leftmost column is original images. The next 9 columns show examples of targeted attacks against nine classes other than the original image class without undergoing perturbation pruning.

**Figure 4 sensors-20-07158-f004:**
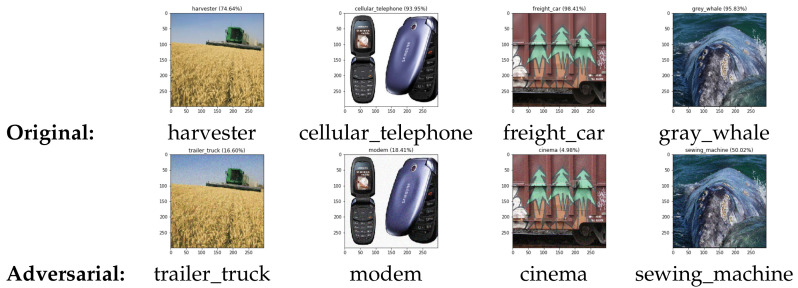
Our adversarial examples on ImageNet with *B* set to 0.05.

**Figure 5 sensors-20-07158-f005:**
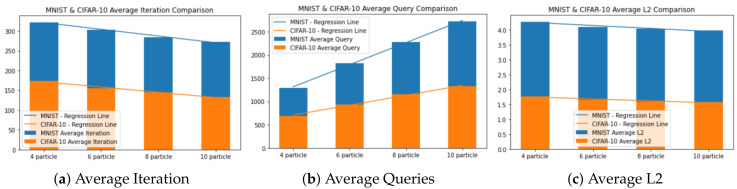
Particle size evaluation. *X*-axis for all graphs represents the particle size (from 4 to 10). *Y*-axis for (**a**) is average iteration. *Y*-axis for (**b**) is average queries. *Y*-axis for (**c**) is average $L_{2}$ distances.

**Figure 6 sensors-20-07158-f006:**
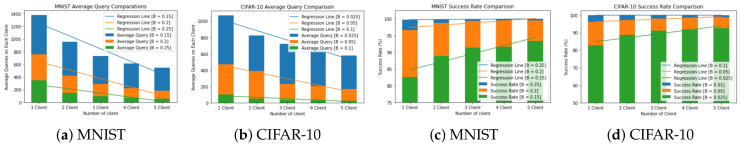
Distributed Approach and Constrained Boundary Evaluation. *X*-axis for all graphs are number of client. *Y*-axis for (**a**),(**b**) are required average queries, while (**c**),(**d**) are the success rate. For MNIST, the blue, orange, and green bars are the results when the constrained boundary is set to 0.25, 0.2, and 0.15, respectively. For CIFAR-10, the blue, orange, and green bars are the results when constrained boundary set to 0.1, 0.05, and 0.025, respectively. Using this distributed approach, the number of query can be suppressed, even to reach lower than 400 while maintaining 100% attack success rate (blue bars).

**Figure 7 sensors-20-07158-f007:**
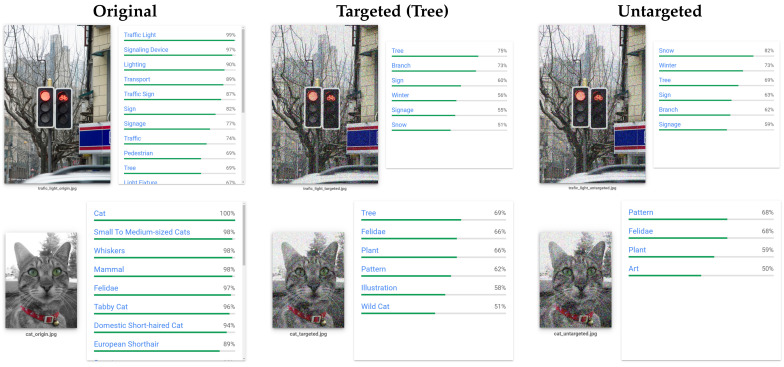
Our adversarial examples on Google Cloud Vision.

**Table 1 sensors-20-07158-t001:** Attack performance comparison using C&W model on MNIST and CIFAR-10 datasets.

MNIST						
	Untargeted			Targeted		
Attack	Success Rate	Avg. $L_{2}$	Avg. Queries	Success Rate	Avg. $L_{2}$	Avg. Queries
ZOO	**100%**	1.4955	384,000	98.90%	**1.9871**	384,000
C&W (Black-box)	33.3%	3.611	4650	26.74%	5.272	4650
AdversarialPSO	96.30%	4.1431	**593**	72.57%	4.7780	1882
SWISS	**100%**	3.4298	3043	19.41%	3.5916	20026
MGRR-PSO Attack	**100%**	4.2805	1288	**100%**	6.0341	**1344**
**CIFAR-10**						
	Untargeted			Targeted		
Attack	Success Rate	Avg. $L_{2}$	Avg. Queries	Success Rate	Avg. $L_{2}$	Avg. Queries
ZOO	**100%**	**0.1997**	128,000	96.80%	**0.3988**	128,000
C&W (Black-box)	25.3%	2.9708	4650	5.3%	5.7439	4650
AdversarialPSO	99.60%	1.4140	1224	71.97%	2.9250	6512
SWISS	99.8%	2.3248	2457	31.93%	2.9972	45308
MGRR-PSO Attack	**100%**	1.767	**694**	**100%**	3.6315	**1071**

**Table 2 sensors-20-07158-t002:** Untargeted attack performance for inception-v3 network on ImageNet dataset.

ImageNet (Untargeted)		
Success Rate	Max. $L_{\infty }$	Avg. Queries
89%	0.05	2472

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
