# Peer review of "A Distributed Black-Box Adversarial Attack Based on Multi-Group Particle Swarm Optimization"

_sensors, 2020, doi:10.3390/s20247158_

Round 1
Reviewer 1 Report
The paper deals with optimization within a very modern topic taken in part from game theory, where the essence is teaching using adverse examples (unfavorable examples). Adverse examples are often discussed during this period in many domains of artificial intelligence. I did not find significant errors in the manuscript, even the symbolic mathematical parts are written elegantly and carefully. I think the overall writing is balanced. This study was carried out on standardised databases of images for a sufficient amount of cases and showed statistically significant differences.
Remark 1: Clip operator should be better specified. (note: The clip operator may be well known in a particular community, but it is probably not well known.)
Remark 2: A1, A2 are the acceleration coefficients present in the MGRR-PSO Attack algorithm, but are not present in the system of equations (1), (2). (I don't see any added mathematically clear comments to this modification in the main text). In order to illustrate the clear context of the original system, in addition to the pseudo-algorithmic forms (MGRR-PSO attack), some mathematical relationships should be mentioned in the main text which may contribute to the dissemination of the method.
Remark 3: Notation p_0 and p_t (Eqs. 3,4) is very close to the notation p_{gd}, p_{id}.
In my opinion, one of the symbols should be changed more significantly to avoid confusion.
Remark 4: It would be beneficial to add to the comparison or to discuss how fitness, particularly in relation to its metrics (between the predicted label and the original label), can influence the outcomes.
Reviewer 2 Report
The authors propose a novel distributed PSO-based methodology for adversarial attacks in the deep learning context. The main contributions of this work are: a distributed architecture which allows to use more than one machine for the attack, the design of a multi-group variant of the well known PSO metaheuristic, a novel pruning which for removal of unnecessary perturbations.
The article is well written and the proposal is interesting. However, I have some concerns that the authors should address before publication:
- Experimental comparisons does not use at all any statistical test. Nowadays statistical test are necessary in order to validate the significance of the results/comparison performed. I suggest the authors to give a read, for instance, to the tutorial at reference [1] (see below), and then validate their comparisons/conclusions by means of one or more statistical test.
- Metaheuristics such as PSO are mainly based on three compenents: solution representation, objective function, variation operators. Actually, solution representation is only stated at the second line of section 4.2. I think it is too late to understand what the algorithm is optimizing. I suggest to formalize solutions representation before in the discussion.
Minor:
- at line 316, "constant values" <-> "parameter values"
Suggestion:
- I suggest the authors to mention, perhaps in the future work part, the Algebraic PSO algorithm (see reference [2] at the end of this review) which allows to apply the same PSO dynamics to search space not necessarily formed by numerical vectors. This can allow to apply their same technique also to other domains in which bit-strings or other discrete representations such as permutations are required.
[1] Derrac, Joaquín, et al. "A practical tutorial on the use of nonparametric statistical tests as a methodology for comparing evolutionary and swarm intelligence algorithms." Swarm and Evolutionary Computation 1.1 (2011): 3-18.
[2] Marco Baioletti, et al. "Tackling permutation-based optimization problems with an algebraic particle swarm optimization algorithm." Fundamenta Informaticae 167.1-2 (2019): 133-158.
References:
[1]
Round 2
Reviewer 2 Report
I am satisfied with authors' revision.